# LADDER POLYNOMIAL NEURAL NETWORKS

## ABSTRACT

Polynomial neural networks are essentially polynomial functions. These networks are shown to have nice theoretical properties by previous analysis, but they are hard to train when their polynomial orders are high. In this work, we devise a new type of activations and then create the Ladder Polynomial Neural Network (LPNN). The LPNN has a feedforward structure. It provides good control of its polynomial order because its order increases by 1 with each of its hidden layers. The new network can be treated with deep learning techniques such as batch normalization and dropout. As a result, it can be well trained with generic optimization algorithms regardless of its depth. In our empirical study, deep LPNN models achieve good performances in a series of regression and classification tasks.

## 1 INTRODUCTION

A polynomial neural network has a unique underlying function, which is a polynomial. Well studied by mathematicians, polynomial functions have been shown to have many favorable theoretical properties. Polynomial neural networks also bridge the analysis of general neural architectures to the properties of polynomial functions. However, in practice, there is still a gap in performance between polynomial neural networks and popular feedforward neural networks. One purpose of this work is to narrow the gap and provide a new model for both theoretical study and practice.

One method of constructing polynomial neural networks is to use the quadratic function as activations in a feedforward network. Livni et al. (2014) call this specific type of networks as PNNs and show several of their theoretical properties, e.g. they are polynomial-time learnable, and they can approximate other feedforward neural networks. Kileel et al. (2019) study their functional space from the algebraic perspective. The theoretical analysis favors low-order polynomials in terms of learning time. However, PNNs cannot have an arbitrary order, as their orders grow exponentially with their layers. For learning PNN models, researchers often devise specialized learning algorithms (Livni et al., 2014; Soltani & Hegde, 2018; Du & Lee, 2018; Soltani & Hegde, 2019). Most of these algorithms only work for PNNs with one or two hidden layers. A deep PNN will have a very large order, then the model becomes hard to train.

Polynomial learning models can also be constructed by defining a polynomial kernel over input features and network parameters (Blondel et al., 2016b;a; 2017). Blondel et al. (2016b) show that PNNs with one hidden layer can be constructed by a second-order polynomial kernel. One can use an arbitrary order in the polynomial kernel, then the constructed learning model has the same order. Factorization machines (Rendle, 2010; Blondel et al., 2016a) is another polynomial model constructed from the ANOVA kernel. Training of these models often needs special optimization methods. They don't have a layered structure, so it is not easy to apply deep learning techniques, such as batch normalization, to these models.

In this work, we propose a new type of polynomial neural networks, Ladder Polynomial Neural Networks (LPNNs). Particularly, we introduce a new activation function, *product activation*, and use it in a feedforward structure. In this new activation function, the hidden layer is multiplied to a linear transformation of the input feature. This activation always increases the polynomial order of the network by one, so we can specify an arbitrary order for the network. As a type of feedforward networks, an LPNN can be trained with standard training techniques. Further analysis shows that the function surface of LPNN has much better smooth guarantee than PNNs. As a function of learnable parameters, the network is block multiconvex with respect it weight

matrices, providing opportunities of faster learning methods. The model space of LPNNs includes models constructed from polynomial kernels as special cases, indicating LPNNs are more flexible than these models.

The LPNN model is evaluated in a list of benchmark regression and classification tasks. The results indicate that the LPNN outperforms other polynomial networks in general. Its classification performances match those of a standard feedforward neural network. Our results also indicate the necessity of batch normalization and dropout when training deep LPNN models.

## 2 THE LADDER POLYNOMIAL NEURAL NETWORK

We first define the general form of a feedforward neural network. Suppose the input to the neural network is a feature vector $\mathbf{x} \in \mathbb{R}^{d_0}$, and denote $\mathbf{h}^0 = \mathbf{x}$. Suppose the network has $L$ hidden layers, with each layer $\ell \in \{1, \ldots, L\}$ takes the input $\mathbf{h}^{\ell-1}$ and has the output $\mathbf{h}^\ell$. Each layer is defined by

$$\mathbf{h}^\ell = \sigma\left(\mathbf{W}^\ell \mathbf{h}^{\ell-1}\right). \tag{1}$$

Here $\mathbf{W}^\ell$ is the weight matrix for layer $\ell$, and $\sigma(\cdot)$ is the activation function. Note that superscripts of hidden vectors and weight matrices always denote layer indices, not exponents. For notational simplicity, we omit intercept vectors for now and will add them back later.

We first define the *product activation* $\sigma_p(\cdot)$ before we define the new network,

$$\sigma_p(\mathbf{u}; \mathbf{V}, \mathbf{x}) = \mathbf{u} \odot (\mathbf{V}\mathbf{x}). \tag{2}$$

Here $\odot$ is the element-wise product. $\mathbf{V}$ is the parameter for the activation function. $\mathbf{V}$ has a shape of $(d, d_0)$ if $\mathbf{u}$ has $d$ entries. The activation is data-dependent. If $\mathbf{u}$ is a polynomial function of $\mathbf{x}$, then $\sigma_p(\mathbf{u}; \mathbf{V}, \mathbf{x})$ is also a polynomial function of $\mathbf{x}$ with the polynomial order increased by 1. Note that the product activation is not a function of $\mathbf{u}$ because different $\mathbf{x}$ values may give the same $\mathbf{u}$ value but different responses from $\sigma_p(\mathbf{u}; \mathbf{V}, \mathbf{x})$.

We use product activations in a feedforward structure and get an LPNN. We use a different matrix $\mathbf{V}^\ell$ for the product activation in each layer $\ell$. Suppose $\mathbf{h}_L$ is the output of the neural network, the function of the LPNN is formally defined as $\mathbf{h}_L = LPNN(\mathbf{x}; \theta)$,

$$LPNN(\mathbf{x}; \theta) := \sigma_p\left(\mathbf{W}^L \sigma_p\left(\mathbf{W}^{L-1} \ldots \sigma_p\left(\mathbf{W}^1\mathbf{x}; \mathbf{V}^1, \mathbf{x}\right) \ldots; \mathbf{V}^{L-1}, \mathbf{x}\right); \mathbf{V}^L, \mathbf{x}\right). \tag{3}$$

Here we use $\theta$ to denote all network parameters, $\theta = \left(\mathbf{W}^1, \ldots, \mathbf{W}^L, \mathbf{V}^1, \ldots, \mathbf{V}^L\right)$. The first hidden layer $\mathbf{h}^1$ is a second order polynomial of the input, and each activation increase the order by 1, so the hidden layer $\mathbf{h}^\ell$ is an order $(\ell+1)$ polynomial. The entire network is an order $L+1$ polynomial function of the input.

We further re-write the function with simple additions and multiplications.

$$\mathbf{h}^\ell = \left(\mathbf{W}^\ell \mathbf{h}^{\ell-1}\right) \odot \left(\mathbf{V}^\ell \mathbf{x}\right) = \sum_{i=1}^{d_\ell} \mathbf{e}_i \left(\mathbf{W}_i^\ell \mathbf{h}^\ell\right)\left(\mathbf{V}_i^\ell \mathbf{x}\right). \tag{4}$$

Here $\mathbf{e}_i$ is a one-hot vector with its $i$-th entry to be one. Note that $\mathbf{W}_i^\ell$ and $\mathbf{V}_i^\ell$ are both row vectors, so both $\left(\mathbf{W}_i^\ell \mathbf{h}^\ell\right)$ and $\left(\mathbf{V}_i^\ell \mathbf{x}\right)$ are scalars.

After expanding all product activations and taking out all summations, we write the entire network as

$$LPNN(\mathbf{x}; \theta) = \sum_{i_L=1}^{d_L} \mathbf{e}_{i_L} \sum_{i_{(L-1)}=1}^{d_{(L-1)}} \cdots \sum_{i_1=1}^{d_1} \left[\left(\prod_{\ell=2}^{L} \mathbf{W}_{i_\ell,i_{\ell-1}}^\ell\right)\left(\mathbf{W}_{i_1}^\ell \mathbf{x}\right)\left(\prod_{\ell=1}^{L} \hat{\mathbf{V}}_{i_\ell,:}^\ell \hat{\mathbf{x}}\right)\right]. \tag{5}$$

This equation further show that the polynomial order increases with the number of layers.

In this form, every monomial term has order $L+1$. To include terms with different orders, we need to include intercept vectors. Suppose each layer has an intercept vector $\mathbf{b}^\ell$, then the layer is defined by

$$\mathbf{h}^\ell = \sigma_p(\mathbf{W}^\ell \mathbf{h}^{\ell-1} + \mathbf{b}^\ell; \mathbf{V}^\ell, \mathbf{x}). \tag{6}$$

We can re-write the input, hidden vectors, and weight matrices in the following form,

$$\hat{\mathbf{x}} = \begin{bmatrix} \mathbf{x} \\ 1 \end{bmatrix}, \quad \hat{\mathbf{h}}^\ell = \begin{bmatrix} \mathbf{h} \\ 1 \end{bmatrix}, \quad \hat{\mathbf{W}}^\ell = \begin{bmatrix} \mathbf{W}^\ell & \mathbf{b}^\ell \\ \mathbf{0}^\top & 1 \end{bmatrix}, \quad \hat{\mathbf{V}}^\ell = \begin{bmatrix} \mathbf{V}^\ell & \mathbf{0} \\ \mathbf{0}^\top & 1 \end{bmatrix}, \qquad (7)$$

then we still have the previous form, $\hat{\mathbf{h}}^\ell = \sigma_p(\hat{\mathbf{W}}^\ell \hat{\mathbf{h}}^{\ell-1}; \hat{\mathbf{V}}^\ell, \hat{\mathbf{x}})$, and then all previous derivations apply. For notational simplicity, the following analysis continues to use notations without the intercept term.

## 3 ANALYSIS

### 3.1 SMOOTHNESS OF THE FUNCTION SURFACE

The smoothness of the function surface of a neural network characterizes the network's important properties, such as being robust to input perturbations. This subsection gives an upper bound of the Lipschitz constant of the LPNN function.

We first compute the first order derivative of $\mathbf{h}^\ell(x)$ with respective to the input $\mathbf{x}$. According to (4), we have,

$$\nabla \mathbf{h}^1 = \text{diag}(\mathbf{V}^1 \mathbf{x})\mathbf{W}^1 + \text{diag}(\mathbf{W}^1 \mathbf{x})\mathbf{V}^1, \tag{8}$$

$$\nabla \mathbf{h}^\ell = \text{diag}(\mathbf{V}^\ell \mathbf{x})\mathbf{W}^\ell(\nabla \mathbf{h}^{\ell-1}) + \text{diag}(\mathbf{W}^\ell \mathbf{h}^{\ell-1})\mathbf{V}^\ell, \quad \ell = 2, 3, \cdots, L \tag{9}$$

where $\text{diag}(\mathbf{u})$ denotes a diagonal matrix with $\mathbf{u}$ as its diagonal.

Then we can give a bound of the first order derivative, which in turn gives an estimate of the Lipschitz constant of $\mathbf{h}^\ell$ as a function of $\mathbf{x}$. The results are summarized in the following theorem. Here, for the sake of simplicity, we use $l^2$ norm ($\|\cdot\|$) for both vectors and matrices.

**Theorem 3.1.** *If $\mathbf{h}^\ell$ is defined recursively as in* (1) *and* (2)*, then, for $\ell = 1, 2, \ldots, L$,*

$$\|\mathbf{h}^\ell\| \leq \left( \prod_{k=1}^\ell \|\mathbf{V}^k\| \|\mathbf{W}^k\| \right) \|\mathbf{x}\|^{\ell+1} = \left( \prod_{k=1}^\ell \rho(\mathbf{V}^k)\rho(\mathbf{W}^k) \right) \|\mathbf{x}\|^{\ell+1} \tag{10}$$

*and*

$$\|\nabla \mathbf{h}^\ell(\mathbf{x})\| \leq (\ell+1) \left( \prod_{k=1}^\ell \|\mathbf{V}^k\| \|\mathbf{W}^k\| \right) \|\mathbf{x}\|^\ell = (\ell+1) \left( \prod_{k=1}^\ell \rho(\mathbf{V}^k)\rho(\mathbf{W}^k) \right) \|\mathbf{x}\|^\ell, \tag{11}$$

*where $\rho(\cdot)$ takes the maximal singular value of its input matrix.*

*Proof.* Our proof is based on two inequalities. The first one is from the definition of matrix norm, e.g. $\|\mathbf{W}^\ell \mathbf{h}^{\ell-1}\| \leq \rho(\mathbf{W}^\ell) \|\mathbf{h}^{\ell-1}\|$. The second one is the Cauchy-Schwarz inequality, e.g. $\|(\mathbf{W}^\ell \mathbf{h}^{\ell-1}) \odot (\mathbf{V}^\ell \mathbf{x})\| \leq \|(\mathbf{W}^\ell \mathbf{h}^{\ell-1})\| \cdot \|(\mathbf{V}^\ell \mathbf{x})\|$.

We will use mathematical induction to derive the results. For $\ell = 1$,

$$\|\mathbf{h}^1\| = \|\text{diag}(\mathbf{V}^1 \mathbf{x})\mathbf{W}^1 \mathbf{x}\| \leq \|\mathbf{V}^1\| \|\mathbf{W}^1\| \|\mathbf{x}\|^2,$$

$$\|\nabla \mathbf{h}^1\| = \|\text{diag}(\mathbf{V}^1 \mathbf{x})\mathbf{W}^1 + \text{diag}(\mathbf{W}^1 \mathbf{x})\mathbf{V}^1\| \leq 2\|\mathbf{V}^1\| \|\mathbf{W}^1\| \|\mathbf{x}\|.$$

This means (10) and (11) hold for $\ell = 1$. Now assume (10) and (11) hold for $\ell - 1$, then

$$\|\mathbf{h}^\ell\| = \|\text{diag}(\mathbf{V}^\ell \mathbf{x})\mathbf{W}^\ell h^{\ell-1}\| \leq \|\mathbf{V}^\ell\| \|\mathbf{x}\| \|\mathbf{W}^\ell\| \|\mathbf{h}^{\ell-1}\| \leq \left( \prod_{\ell=1}^k \|\mathbf{V}^\ell\| \|\mathbf{W}^\ell\| \right) \|\mathbf{x}\|^{\ell+1},$$

and

$$\|\nabla \mathbf{h}^\ell\| = \| \text{diag}(\mathbf{V}^\ell \mathbf{x})\mathbf{W}^\ell(\nabla \mathbf{h}^{\ell-1}(\mathbf{x})) + \text{diag}(\mathbf{W}^\ell \mathbf{h}^{\ell-1}(\mathbf{x}))\mathbf{V}^\ell\|$$
$$\leq \|\mathbf{V}^\ell\| \|\mathbf{x}\| \|\mathbf{W}^\ell\| \|\nabla \mathbf{h}^{\ell-1}(\mathbf{x})\| + \|\mathbf{W}^\ell\| \|\mathbf{h}^{\ell-1}(\mathbf{x})\| \|\mathbf{V}^\ell\|$$
$$= (\ell+1) \left( \prod_{k=1}^\ell \|\mathbf{V}^k\| \|\mathbf{W}^k\| \right) \|\mathbf{x}\|^\ell.$$

$\square$

This bound indicates the relation between the smoothness and the norm of weight matrices. The result is similar to the feedforward neural networks with other activation functions whose gradients are always between -1 and 1 (Virmaux & Scaman, 2018). The proof above is independent of the data and network parameters. We can further improve the bound by considering network weights with an approach similar to (Virmaux & Scaman, 2018), but we defer the further investigation to future work.

As a comparison, the same type of bound is much higher for a deep PNN model, as its order is exponential in its number of layers.

## 3.2 MULTICONVEXITY IN PARAMETERS

We also study the network as a function of its parameters to understand its function surface for parameter optimization.

The network output $\mathbf{h}_L$ is linear in a weight matrix $\mathbf{W}^\ell$ or $\mathbf{V}^\ell$ if we hold other weights as constants. We can see so if we examine the function form in (5): there is no multiplications between any two entries in $\mathbf{W}^\ell$ or $\mathbf{V}^\ell$. With this property, the learning model is block multiconvex with respect to the model parameters.

**Theorem 3.2.** *Let* $\mathbf{h}_L = LPNN(\mathbf{x}; \theta)$ *and* $\mathbf{y}$ *be the fitting target. Suppose* $\mathrm{loss}(\mathbf{h}_L, \mathbf{y})$ *is a convex loss function, then the training objective of an LPNN,*

$$obj(\theta) = \mathrm{loss}(\mathbf{h}_L, \mathbf{y}) + \frac{1}{2}\lambda \sum_{\ell=1}^{L} \left( \|\mathbf{W}^\ell\|_F^2 + \|\mathbf{V}^\ell\|_F^2 \right), \tag{12}$$

*is block multiconvex in blocks* $\{\mathbf{W}^1, \ldots, \mathbf{W}^L, \mathbf{V}^1, \ldots, \mathbf{V}^L\}$.

Multiconvexity enables some specialized optimization algorithms (Shen et al., 2017). These algorithms have the potential to be combined with generic optimization algorithms and speed up the training of LPNN.

As a comparison, network weights in PNNs are exponentiated through hidden layers. We believe PNNs has a much rough surface than LPNNs.

## 3.3 TRAINING WITH BATCH NORMALIZATION AND DROPOUT

Batch normalization (BN) (Ioffe & Szegedy, 2015) and dropout (Srivastava et al., 2014) are effective techniques for training deep neural networks. We can apply batch normalization and dropout to an LPNN without any modification given its layered structure. We put the BN layer after the activation per some practitioners' advice, though the original paper suggests putting it before the activation.

When LPNN has BN layers, the model in training is not a polynomial function, but the trained model with constant BN parameters are still polynomial functions. Here we want to integrate BN parameters into network weights so that previous derivations still apply. We consider one hidden layer and omit layer indices for notational simplicity.

Let $[\mathbf{h}_1, \ldots, \mathbf{h}_n]$ be hidden layer values of $n$ instances in a batch, then the batch-normalized hidden layer $\bar{\mathbf{h}}_i$ of instance $i$ is computed by

$$\bar{\mathbf{h}}_i = \gamma(\mathbf{h}_i - \boldsymbol{\mu})/(\boldsymbol{\sigma} + \epsilon) + \beta, \ \ \text{with} \ \ \boldsymbol{\mu} = \frac{1}{n}\sum_{i=1}^{n} \mathbf{h}_i, \ \boldsymbol{\sigma} = \frac{1}{n}\sum_{i=1}^{n}(\mathbf{h}_i - \boldsymbol{\mu})^2. \tag{13}$$

Here the division / and the square are element-wise operations, and $\epsilon$ is a small positive number.

In the training procedure, the variance $\boldsymbol{\sigma}$ is a function of input instances in the batch, therefore, the hidden vector $\bar{\mathbf{h}}_i$ is not a linear function of the input $\mathbf{x}$ anymore. After training, the mean vector $\boldsymbol{\mu}$ and the variance vector $\boldsymbol{\sigma}$ become constants, then the model is still a polynomial function.

Let's write the model with BN into the original function form. Let $\boldsymbol{\mu}, \boldsymbol{\sigma}, \gamma$, and $\beta$ are convergent values of the training procedure, then the equivalent LPNN without batch normalization has parameters defined as follows.

$$\mathbf{W}' = \mathbf{W} \operatorname{diag}(\gamma/(\boldsymbol{\sigma} + \epsilon)), \quad \mathbf{b}' = -\mathbf{W} \operatorname{diag}(\gamma\boldsymbol{\mu}/(\boldsymbol{\sigma} + \epsilon) + \beta \tag{14}$$

Here we need to build equivalence with the LPNN with intercept terms. These terms can be absorbed into weight matrices by (7).

In this result, we can see that BN changes the norm of weight matrices and then the Lipschitz constant of the network function. Based on the study by Santurkar et al. (2018), BN can simply shrink the norms of weight matrices to avoid having steep slopes in the function surface.

Dropout can be directly applied to LPNN. In the training phase, using dropout is equivalent to removing some entries in summations of (5) and rescaling the summation. In the testing phase, dropout have no effect, and the trained model is just as the definition above.

### 3.4 Relation with models based on polynomial kernels

In this subsection, we show that the polynomial networks constructed from polynomial kernel functions (Blondel et al., 2016b) are special cases of LPNN networks.

**Lemma 3.3.** *The polynomial kernel function $\mathcal{P}^m(\mathbf{p}, \mathbf{x}) = (\lambda + \mathbf{p}^\top \mathbf{x})^m$ with $\mathbf{p}, \mathbf{x} \in \mathcal{R}^d$ can be written in the form of $LPNN(\mathbf{x}; \theta)$ such that network weights in $\theta$ can be expressed by $\mathbf{p}$.*

*Proof.* Append 1 to the feature vector, $\mathbf{h}_0 = [\mathbf{x}^\top, 1]^\top$. Set $\mathbf{W}^1 = [\mathbf{p}^\top, \lambda]^\top$, $\mathbf{V}^\ell = [\mathbf{p}^\top, \lambda]^\top$ for all $\ell = 1, \ldots, m-1$, and $\mathbf{W}_{ll} = [1]$ for $\ell = 2, \ldots, L$, then $LPNN(\mathbf{x}; \theta)$ is equivalent to the kernel by (5). $\square$

**Theorem 3.4.** *The learning models in the form of $y = \sum_{k=1}^K \pi_k (\lambda + \mathbf{p}_k^\top \mathbf{x})^m$ (Blondel et al., 2016b) are special cases of LPNN.*

**Theorem 3.5.** *The second order factorization machines are special cases of LPNN.*

*Proof.* By Appendix D.3 in (Blondel et al., 2016b) , the function of a factorization machine can be computed by

$$FM(\mathbf{x}; \mathbf{U}, \mathbf{S}) = \frac{1}{2} \left[ \mathbf{x}^\top \mathbf{U} \mathbf{S}^\top \mathbf{x} - \operatorname{tr}\left( \mathbf{U}^\top \operatorname{diag}(\mathbf{x}) \operatorname{diag}(\mathbf{x}) \mathbf{S} \right) \right]$$
$$= \frac{1}{2} \left[ \mathbf{x}^\top \mathbf{U} \mathbf{S}^\top \mathbf{x} - (\mathbf{x} \odot \mathbf{x})^\top \operatorname{diag}\left( \mathbf{S} \mathbf{U}^\top \right) \right] \tag{15}$$

We can set the network as follows.

$$\mathbf{W}^1 = \left[ \begin{array}{c} \mathbf{U} \\ \mathbf{I} \end{array} \right], \mathbf{V}^1 = \left[ \begin{array}{c} \mathbf{S} \\ \mathbf{I} \end{array} \right] \tag{16}$$

Here the identity matrix in $\mathbf{W}^1$ and $\mathbf{V}^1$ passes $\mathbf{x}$ to the hidden layer and get $(\mathbf{x} \odot \mathbf{x})$ there. The hidden layer $\mathbf{h}^1 = [(\mathbf{U}\mathbf{x}) \odot (\mathbf{S}\mathbf{x}); \mathbf{x} \odot \mathbf{x}]$. Then we set $\mathbf{W}^2 = [\mathbf{1}^\top, \mathbf{t}^\top]$ with $\mathbf{t} = \operatorname{diag}\left( \mathbf{S} \mathbf{U}^\top \right)$, then $\mathbf{W}^2 \mathbf{h}^1 = FM(\mathbf{x}; \mathbf{U}, \mathbf{S})$. $\square$

From this analysis, we see that the LPNN is more flexible than polynomial models constructed from polynomial kernel functions. We believe the LPNN is also more flexible than the factorization machine. Compared with LPNNs, the factorization machine does not have monomials that contains the second or higher order exponential of a feature entry. We don't expect this difference brings much expressiveness to the factorization machine, though it creates difficulties for writing a factorization machine in the form of an LPNN.

**Table 1:** RMSE of different models on regression tasks

| methods | wine-quality | power-plant | kin8nm | boston-housing | concrete-strength |
|---------|--------------|-------------|--------|----------------|-------------------|
| FF | 0.60 ± 0.04* | 4.05 ±0.17* | 0.100 ± 0.002 | 2.82 ± 0.76* | 5.10 ±0.49* |
| FM | 0.73 ± 0.09 | 4.43 ± 0.15 | 0.155 ± 0.004 | 4.80 ± 1.14 | 8.52 ±0.59 |
| PK | 4.39 ± 5.50 | 4.05 ± 0.15* | 0.100 ± 0.005 | 41.9 ± 77.2 | 7.95 ±2.42 |
| PNN | 5.49 ± 16.5 | 5.83 ± 1.39 | 0.102 ± 0.007 | 4.59 ± 2.74 | 5.58 ±0.48 |
| LPNN | 0.82± 0.18 | 4.13 ± 0.16 | 0.099 ± 0.006* | 4.05 ± 2.13 | 5.20 ±0.62 |

**Table 2:** Error rates of different models on classification tasks

| methods | mnist | fashion-mnist | skin | sensIT | letter | covtype-b | covtype |
|---------|-------|---------------|------|--------|--------|-----------|---------|
| FF | 0.0185 | **0.108** | 0.0313 | 0.176 | 0.096 | 0.113 | 0.146 |
| FM | 0.0573 | 0.167 | 0.0439 | 0.260 | 0.546 | 0.208 | 0.575 |
| PK | 0.0506 | 0.168 | 0.0039 | 0.225 | 0.248 | 0.191 | 0.494 |
| PNN | 0.0503 | 0.127 | 0.0018 | 0.199 | 0.104 | **0.097** | **0.103** |
| LPNN | **0.0171** | 0.117 | **0.0017** | **0.175** | **0.0729** | 0.117 | 0.140 |

# 4 EXPERIMENT

## 4.1 EXPERIMENT SETUP

In this section, we evaluate the LPNN on several learning tasks. The LPNN is compared against feedforwrad network and three polynomial learning models. All models are summarized below.

*Feedforward network (FF):* FF uses ReLU functions as activations. We add $l$-2 norm regulraization to the model. The regularization weight is chosen from {1e-6, 1e-5, 1e-4, 5e-4}. When dropout is applied, the dropout rate is chosen from {0, 0.05, 0.1, 0.2, 0.4}.

*Polynomial Neural Network (PNN):* the model is the same as the FF except its activations are the quadratic function. It has the same hyperparameters as FF, and it is trained in the same way as FF.

*Factorization Machine (FM):* we use the implementation from the `sklearn` package (Niculae, Accessed in 2019). The order of FM in this implementation can be 2 or 3. It add several ANOVA kernel functions (called factors) to increase model complexity. The model is also regularized by $l$-2 norm. The hyperparameters of FM include the order, the number of factors, and the weight of regularization. The number of factors is chosen from from {2, 4, 8, 16}, and the regularization weight is chose from the same range as FF. This implementation of FF does not have multiple outputs, so we have used one-vs-rest for multiclass classification problems.

*Polynomial Kernel (PK):* PK uses polynomial kernels. Other than that, PK is similar to FM. We can specify the order of the underlying polynomial function of PK. The hyperparameters of PK are the same as FM. The implementation is also from the `sklearn` package.

*LPNN:* the model is the same as the FF except its activations are product activations. Its hyperparameters are the same as FF, and it is trained in the same way as FF.

We test these models on five regression datasets (wine-quality, power-plant, kin8nm, boston-housing, and concrete-strength) and six classification datasets (mnist, fashion-mnist, skin, sensIT, letter, covtype-b, and covtype). The mnist and fasion-mnist datasets come with the Keras package, the skin, sensIT, and covtype-b datasets are from the libSVM website, and all other datasets are from the UCI repository.

## 4.2 PRODUCT ACTIVATIONS

We first examine the product activation function $\mathbf{h} = \sigma_p(\mathbf{u}; \mathbf{V}, \mathbf{x})$ in a trained model. We set up an LPNN with three hidden layers and then train it on the mnist dataset. The training finishes after 20 epoches when the network has a validation accuracy of 0.984. Then we check inputs

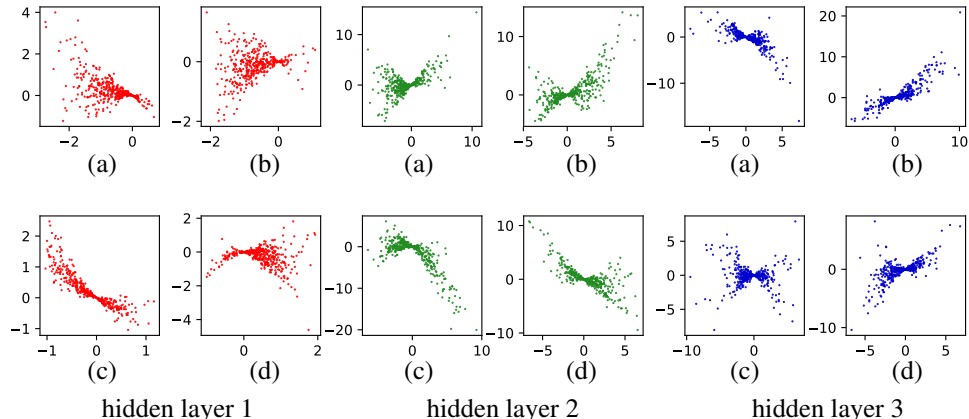

**Figure 1:** Product activations of LPNN on the mnist dataset. The model has three hidden layers. From each layer, activations of four hidden units are plot here in the same color.

**Table 3:** Effect of batch normalization and dropout

| $L$ | BN and dropout | only dropout | only BN | neither |
|---|---|---|---|---|
| 1 | $7.77 \pm 0.53$ | $7.78 \pm 0.54$ | $7.82 \pm 0.55$ | $7.76 \pm 0.53$ |
| 2 | $6.05 \pm 0.50$ | $5.98 \pm 0.57$ | $6.26 \pm 0.87$ | $6.30 \pm 0.96$ |
| 3 | $5.20 \pm 0.62$ | $5.12 \pm 0.58$ | $5.82 \pm 1.13$ | $6.89 \pm 2.06$ |
| 5 | $4.72 \pm 0.66$ | $4.92 \pm 0.78$ | $5.46 \pm 1.83$ | $7.86 \pm 3.10$ |
| 10 | $5.11 \pm 2.18$ | $4.71 \pm 0.90$ | $4.97 \pm 0.97$ | $7.49 \pm 2.77$ |

and responses of the activation functions at three different layers. We plot the response $h_i$ against the corresponding $u_i$ for each hidden unit $i$ to generate a subplot. We randomly select 400 instances and plot each $(h_i, u_i)$ pair. We plot four hidden units at each of all three hidden layers and generate plots in Figure 1.

In these results, we see that the product activation is not really a function because inputs with the same value may have different responses. It is also clear that the product activation is not linear. The behavior of the activation is versatile: the activations shown in (c) at layer 1 and (b) at layer 3 exhibit some linear behavior while the activations shown in (d) at layer 1 and (c) at layer 2 roughly approximate the quadratic function.

## 4.3 REGRESSION AND CLASSIFICATION

We first apply the model to five regression tasks. We use the same data splits by Gal (Accessed in 2019). Each dataset has 20 random splits. On each split, we run model selection through five-fold cross validation, re-train the model, and then test the model on the test set. The results are averaged over the 20 splits. For FF, PNN, and LPNN, we set three hidden layers and 50 hidden units in each hidden layer. We apply dropout and batch normalization to all the three models. We set the polynomial order to be 4 for the PK model to match the order as LPNN. We set the order of FM to be 3. For each model, we select all hyperparameters described in the subsection above.

Table 1 tabulates RMSE of all algorithms on all datasets. On most datasets, no single algorithm outperforms others with statistical significance, but we put a * at the smallest error mean. If comparing the mean values only, LPNN performs a little worse than FF but better than other polynomial models. The PNN has very bad performances on two splits of the wine-quality dataset. We speculate that PNN is not stable when its polynomial order is high. PK has bad

**Table 4:** Effect of batch normalization and dropout on the mnist dataset

| $L$ | BN and dropout | only dropout | only BN | neither |
|---|---|---|---|---|
| 1 | 0.0191 | 0.0208 | 0.0242 | 0.0171 |
| 2 | 0.0191 | 0.0188 | 0.0202 | 0.0192 |
| 3 | 0.0170 | 0.0187 | 0.0229 | 0.0207 |
| 5 | 0.0207 | 0.7657 | 0.0271 | 0.8947 |
| 10 | 0.0230 | 0.0298 | 0.0207 | 0.0241 |

performances on wine-quality and boston-housing because the model does not fit the two tasks–its performances are bad on most splits.

We then test these models on seven classification tasks. For each dataset, we set 30% as the test set, except for mnist and fasion-mnist datasets, which come with test sets. We do model selection for both architecture and hyperparameters on 20% of the training set. For neural networks, the number of hidden layers is chosen from $\{1, 2, 4\}$. We shrink the number of hidden units from the bottom to the top. The number of hidden units is computed by $\alpha^\ell(d_{out} - d_{in}) + d_{out}$ so that the number of hidden units in a middle layer is between the input dimension and the output dimension. The shrinking factor $\alpha$ is chose from $\{0.3, 0.5, 0.7, 0.8\}$. We also select the order for PK from $\{2, 3, 5\}$ to match the order of LPNN. All other hyperparameters of a model are also selcted together with architectures.

The error rates of different models are reported in Table 2. In general, the performance of LPNN is comparable to FF and better other polynomial models. Comparing to feedforward networks, LPNN has relatively better performance on classification tasks. We speculate the reason is that an LPNN only needs to decide discrete labels from its outputs in classification tasks while it needs to fit the exact value in regression tasks. LPNN may be not flexible enough for fitting continuous values compared to feedforward networks.

### 4.4 THE EFFECT OF BATCH NORMALIZATION AND DROPOUT

In this subsection, we investigate the effect of batch normalization and dropout on LPNN. We use $L \in \{1, 2, 3, 5, 10\}$. For each depth, we try four configurations: using/not using batch normalization and using/not using dropout. We select other hyperparameters through model selection. We run the experiment on a regression task (concrete-strength) and a classification task (mnist).

The results are shown in Table 3. From this result, we see that both batch normalization and dropout are needed to train a good LPNN model. On the mnist dataset, the LPNN without batch normalization has very bad performance when $L = 5$. Its performance drops sharply after a few epochs. This observation indicates that the LPNN without batch normalization is very unstable due to some bad optimization directions.

## 5 CONCLUSION

In this paper, we have proposed LPNN, a new type of polynomial neural networks that can have an arbitrary polynomial order. The network is based on product activations. A product activation multiplies a linear transformation of the input to the hidden layer to achieve non-linear transformation of the input. The product activation always increases the polynomial order by 1. LPNN uses product activations in a feedforward structure. With modern training techniques, such as dropout and batch normalization, we can well train deep LPNN models. These models achieve competitive performances in our empirical evaluations. The LPNN has the potential to provide new insights to the theoretical study of polynomial learning models and. It is also a valuable learning method in practice.

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
