# OpenReview forum: "Ladder Polynomial Neural Networks"
_ICLR.cc/2020/Conference — Reject_

### Official Review · AnonReviewer3 · 2019-10-22
**Official Blind Review #3**

**Rating:** 3

**Review:**

This work introduces a new polynomial feed-forward neural network called Ladder Polynomial Neural Network (LPNN). Theoretical results show that LPNNs generalize vanilla PNNs and FMs. In the experimental analyses, LPNNs perform similar to the vanilla FMs and PNNs, as well.

- In the statement “V has a shape of (d, d0) if u has d entries”, what do you mean by shape, more precisely?

- x is a feature vector fed to a neural network as an input. I assume that it is given in the input layer l=0, according to the notation. Then, should x^l be used instead of x in equations (4), (5), (6), (9), and the other corresponding statements? Otherwise, is V^l applied to the input vector x at each layer l?

- What do \| \|^2, \| \|^l, \| \|^{l+1} denote?

- How do you define the norm \| \| for matrices, more precisely?

- Please provide standard deviation/variance of classification error of different models in Table 2.

- Please clarify novelty and superiority of the proposed LPNNs compared to the vanilla and state-of-the-art methods in theory and practice. For this purpose, I suggest to further analyze and compare convergence and generalization properties of LPNNs with the sota in theory and practice.

After the rebuttal:

Authors responded some of my questions. However, I still consider that the contribution of the paper is limited, and should be improved for a clear acceptance. Therefore, I keep my rating.

**Experience Assessment:**

I have published in this field for several years.

**Review Assessment: Checking Correctness Of Derivations And Theory:**

I assessed the sensibility of the derivations and theory.

**Review Assessment: Checking Correctness Of Experiments:**

I assessed the sensibility of the experiments.

**Review Assessment: Thoroughness In Paper Reading:**

I read the paper at least twice and used my best judgement in assessing the paper.

---

> ### Author Response · Authors · 2019-11-09
> **Response to Reviewer 3:**
>
> Thank you for your feedback. We address your concerns as follows.
>
> 1. " LPNNs perform similar to the vanilla FMs and PNNs, as well". We politely disagree with this statement. Comparing to FM, our model has better performances on 4 out of 5 regression tasks (comparing the mean error only) and all 7 classification tasks. Comparing to PNN, our model has better performance on all 5 regression tasks  (comparing the mean error only) and 5 out of 7 classification tasks.
>
> 2. shape of matrices: the "shape" a matrix means the *size* of the matrix.
>
> 3. the feature vector x: x is always the feature vector, and there is no x^l. Your latter statement is correct: V^l is always applied to the feature vector x. Using x is one key design of the model: we use the input vector x to create non-linearity as the activation (please see Eq.(2)), so the input x is indeed used in every layer.
>
> 4. vector norm: \|x \| is the norm of x, and \|x \|^l is the l-th exponential of the norm.
>
> 5. matrix norm: the matrix norm is induced by the vector norm (2-norm in our case). We will make this clear in the submission. For your reference, the matrix W has norm, \|W \| = sup { \|Wx \| / \|x\|, x \neq 0}.
>
> 6. standard deviation: we have used one test split from every dataset (some datasets provide the test split). We can estimate the standard deviation of an error rate of a trained model by the Central Limit Theorem:  sqrt((1 - error_rate) * error_rate / test_size). The sizes of test splits from all datasets are over 10,000 except the size of the letter dataset is 4,500, so the standard deviations are all small and often neglectable. The standard deviations of our model are (0.0013, 0.0032, 0.0002, 0.0025, 0.0039, 0.0008, 0.0008). The standard derivations of other methods should be similar.
>
> 7.  novelty and superiority of the proposed LPNN: compared to factorization models (FM and PK), LPNN has the feedforward structure and can be trained with standard techniques (e.g. batch normalization and dropout). LPNN also includes FM and PK as special cases. Compared with PNN, LPNN has a controllable order and can be better trained. It is also a multilinear function while PNN is not.
>
> Finally, we would like to summarize our contribution again. First, we propose the new activation, product activation, which leads to a new polynomial model with the feedforward architecture. Then the new model can be trained with standard training techniques. Second, our method connects the feedforward structure with factorization models. Particularly, our model covers two previous factorization models as special cases.  Third, we have shown a few nice properties of the proposed model: it is multilinear, and its smoothness is similar to standard feedforward neural networks.

---

### Official Review · AnonReviewer2 · 2019-10-23
**Official Blind Review #2**

**Rating:** 6

**Review:**

This paper proposes Ladder Polynomial Neural Networks (LPNNs) that use a new type of activation primitive -- a product activation -- in a feed-forward architecture. Unlike other polynomial architectures that grow in the order exponentially with network depth, the proposed approach gives explicit control over the order and smoothness of the network output and enables training with standard techniques.

The proposed architecture is closely related to a decomposition of a k’th order multivariate polynomial function
[T, x^{\otimes k}] = \lambda^\top (A x \odot A x \odot …  \odot A x)	=  \lambda^\top (A x)^{\odot k}
where T is a symmetric tensor of polynomial coefficients and [\cdot,\cdot] denotes contraction. This is a shallow (one layer architecture) and sometimes referred as a Waring decomposition.

In this paper, the authors propose a specific chain factorization of the polynomial (Eq 5 in the paper), where they write the factors recursively, that they name as a ladder polynomial neural network.

h^\ell = (W_\ell h^{\ell-1} \odot V^{\ell} x)

The ladder architecture is very closely related to tensor trains (https://epubs.siam.org/doi/10.1137/090752286). I found it surprising and somewhat alarming that this literature is not being cited as these methods are also quite well known in deep learning.

I like the smoothness analysis of section 3.1 -- the proof is quite easy to follow and direct. I would be quite surprised if this result would not be known in the literature in some other form but I don’t recall seeing it. On the other hand it seems to be inevitably very loose for a deep ladder network unless the network models the zero function. It would have been a valuable addition to the experimental section, if this bound would have been illustrated numerically on synthetic examples.

In 3.2, The authors say that the objective is multiconvex -- I would argue that it is multilinear (apart from the regularization term, that is later introduces). The observation in 3.3, that batch-normalization or dropout can be used for this model is perhaps tangential to the main argument. These is investigated in the experimental section but I don’t see a clear conclusion. The section in 3.4 must include links to tensor decompositions beyond factorization machines.

Overall, I think the paper has some merit and could be interesting for some readers, despite the fact that the contribution is not very original and the treatment could be improved in many ways.


**Experience Assessment:**

I have published one or two papers in this area.

**Review Assessment: Checking Correctness Of Derivations And Theory:**

I assessed the sensibility of the derivations and theory.

**Review Assessment: Checking Correctness Of Experiments:**

I assessed the sensibility of the experiments.

**Review Assessment: Thoroughness In Paper Reading:**

I read the paper at least twice and used my best judgement in assessing the paper.

---

> ### Author Response · Authors · 2019-11-09
> **Response to Reviewer 2:**
>
> Thank you for your insightful comments. We address your concerns as follows.
>
> 1. Thank you for pointing the tensor train paper to us. We have no intention to omit this citation. We don't view or claim the chain factorization of LPNN as our main contribution. Our first contribution is the new activation, the *product activation*. The factorization of LPNN is a consequence of this activation.  Our second contribution is the connection between the feedforward architecture and the factorization. We show the factorization in the paper to provide readers an understanding besides the understanding from the feedforward structure.
>
> Actually, the tensor train paper provides further support for our work. With the method in our paper, we may have an activation that leads to a model corresponding to the tensor train, then we can learn such a model with standard training techniques (e.g. dropout and batch normalization).
>
> 2. smoothness proof: the bound is consistent with the bound in [1]. However, we need to do the proof again because the new activation does not meet the assumption in [1]. There are some advanced techniques in [1] to further tight the smoothness bound. We will consider applying these techniques to our model.
>
> 3. multiconvex vs multilinear: yes, it is better to say the model is multilinear -- we will correct it.
>
> 4. batch normalization and dropout: we want to make a point that batch normalization and dropout are beneficial for training a polynomial model (equivalently the LPNN factorization). The empirical investigation *does* show the performance improvement from dropout or batch normalization or both. In Table 3 and Table 4, errors in the last column are generally larger than the errors in the first three columns. Without batch normalization, the network sometimes does not converge (the large error rates at col 2 & 4, row 4 of Table 4).
>
> 5. citation: yes, we will include the citation to the tensor train paper.
>
> Finally, we hope our explanations clarify some of your concerns. We would like to politely ask you to reconsider the originality of the paper.
>
>
> Citation:
> 1. Aladin Virmaux and Kevin Scaman. Lipschitz regularity of deep neural networks: analysis and
> efficient estimation. In Advances in Neural Information Processing Systems, pp. 3835–3844,
> 2018.

---

### Official Review · AnonReviewer1 · 2019-10-27
**Official Blind Review #1**

**Rating:** 6

**Review:**

In this paper, the authors a new type of polynomial neural networks LPNN that can have an arbitrary polynomial order. The network has a feedforward structure which provides a good control of its polynomial order. Empirical study shows that deep LPNN models achieve good performances in regression and classification tasks. In general, the paper is clearly written by addressing an interesting problem but I still have several concerns.
1.	The authors are expected to analyze the time complexity in terms of both theoretical and experimental analysis since the time cost the one of the major limitation of PNN.
2.	The experimental section is rather weak since the authors only report results in some simple data. The authors are expected to report more complex tasks to show its effectivenss.
3.	It would be interesting if the authors could show whether the algorithm can integrate with other structure such as convolutional operator to cope with the image classification, and other relevant tasks.


**Experience Assessment:**

I have published in this field for several years.

**Review Assessment: Checking Correctness Of Derivations And Theory:**

I assessed the sensibility of the derivations and theory.

**Review Assessment: Checking Correctness Of Experiments:**

I assessed the sensibility of the experiments.

**Review Assessment: Thoroughness In Paper Reading:**

I read the paper thoroughly.

---

> ### Author Response · Authors · 2019-11-09
> **Response to Reviewer 1:**
>
> Thank you for your summary of our work! We address your concerns as follows.
>
> 1. Yes, we will include time complexity in the next version of the submission. The network has a similar structure to the feedforward neural network, so the analysis is similar. We'd like to include a brief analysis here.  Suppose d is the largest of the number of hidden units and the number of features (d = max(d_0, ..., d_L)), B is the batch size, and M is the number of training iterations. Then in the forward computation, each layer takes time O(d^2) to do the two matrix-vector multiplications and the element-wise product. In the backward propagation, each layer takes time O(d^2) to compute the derivatives with respect to (W, V) and also propagate the derivative to the previous layer. Overall the training time is O(M*B*L*d^2). The test time for a single instance is O(L*d^2).
>
> 2. I guess "complex tasks" in the comment means learning tasks on images, audios, etc. (correct me if I am wrong). It is substantially difficult to devise a *polynomial* model and still match the SOTA performance achieved by CNN or other complex models. In this submission, we want to take the first solid step: making a feedforward network also a polynomial function with a controllable order. Our experiments show that the LPNN matches standard feedforward networks in performance (slightly worse in regression tasks). With this purpose, we have used datasets that are commonly used for benchmarking feedforward neural networks. We would also like to politely point out that papers of baseline methods (FM, PK, and PNN)  all have evaluated their models on "simple" datasets.
>
> 3.  We do have a plan to extend this polynomial learning model to include convolutional operations. However, most existing models with other structures  (e.g. CNNs or RNNs) have non-linear operations. The naive combination will not produce a polynomial model. Note that the polynomial model is one of the main purposes of this submission. We will explore different extensions of the proposed model to the future.
>
> Finally, we want to emphasize a little more of the importance of devising polynomial models. As we have discussed in our introduction section, a polynomial model is an important tool for the theorists to understand non-linear models, so the improvement of polynomial models is well justified.

---

### Decision · Program_Chairs · 2019-12-19

**Decision:**

Reject

**Comment:**

This paper proposes a new type of Polynomial NN called Ladder Polynomial NN (LPNN) which is easy to train with general optimization algorithms and can be combined with techniques like batch normalization and dropout.  Experiments show it works better than FMs with simple classification and regression tasks, but no experiments are done in more complex tasks. All reviewers agree the paper addresses an interesting question and makes some progress but the contribution is limited and there are still many ways to improve.